# Potential Use of PLA-Based Films Loaded with Antioxidant Agents from Spent Coffee Grounds for Preservation of Refrigerated Foods

**DOI:** 10.3390/foods12224167

**Published:** 2023-11-17

**Authors:** Margherita Pettinato, Maria Bolla, Roberta Campardelli, Giuseppe Firpo, Patrizia Perego

**Affiliations:** 1Department of Civil, Chemical and Environmental Engineering, Polytechnic School, University of Genoa, Via All’Opera Pia 15, 16145 Genoa, Italy; margherita.pettinato@unige.it (M.P.); maria.bolla01@edu.unige.it (M.B.); p.perego@unige.it (P.P.); 2Department of Physics, Nanomedicine Laboratory, University of Genoa, Via Dodecaneso 33, 16146 Genoa, Italy

**Keywords:** active food packaging, spent coffee grounds, waste valorization, high-pressure and -temperature extraction, polylactic acid, gas barrier properties, lipid peroxidation inhibition

## Abstract

The aim of this work concerned the production of an active food packaging suitable for refrigerated foods. Polylactic-acid-based films were produced by optimizing the solvent casting technique and testing different loadings of extracts obtained from spent coffee grounds. Indeed, an extract obtained by high-pressure and -temperature extraction (HPTE) and a further purified extract by liquid–liquid extraction (LLE) were separately used as active agents, and the effects on packaging features and active compounds migration were analyzed. The selected active agents showed antioxidant and lipid peroxidation inhibition effects on food simulants (peroxide values of 9.2 ÷ 12.0 meqO_2_/kg extra virgin olive oil), demonstrating the possibility of enhancing food shelf life. In addition, significant effects on the packaging structure due to the presence of the extract were observed, since it can enhance gas barrier properties of the polymer (O_2_ permeability of 1.6 ÷ 1.3 × 10^−9^ cm^2^/s) and confer better processability. In general, the HPTE extract exhibited better performances than the further purified extract, which was due to the presence of a complex pool of antioxidants and the browning effect on the film but a limited loading capacity on the polymer (840 μg _caffeine_/g _PLA_), while higher loading capabilities were enabled using LLE extract.

## 1. Introduction

Refrigerated foods, such as meat, seafood products, fruit, and vegetables, generally present with a short shelf life, given their propensity to perish under oxygen, light, and bacterial spoiling [1,2,3].

Nevertheless, the increasing customer awareness towards the consumption of fresh and healthy food is leading to a growing demand for minimally processed food, requiring innovative storage methods [3].

Packaging is crucial in providing food stability and in preventing mechanical damage, chemical or biochemical changes, and microbiological spoilage [4]. Recently, the passive protection role of packaging against external contaminations and environmental conditions has been reconsidered, trying to improve its properties through the incorporation of active functions, directly interacting with food or its surroundings in order to enhance its shelf life and organoleptic properties [5,6].

Depending on the desired final effects, several active materials were tested in the literature [6,7,8,9,10,11], including antioxidant agents, which can reduce and stabilize reactive free radicals, preserving the food’s nutritional value, its organoleptic properties, and shelf life. New food packaging technologies are in development to enhance the food shelf life, preferentially adopting naturally based active agents, such as extracts from rosemary [12], *Piper betel* leaf [13], thyme [14], potato peels [15], and lemon by-products [16].

Efforts have been made to incorporate into the polymer matrices a wide range of natural and biodegradable additives derived from agro-industrial wastes and plants with a high degree of safety and a low environmental footprint [17,18]. Bio-based extracts with antioxidant and antimicrobial properties were demonstrated to be effective natural additives able to prevent the microbiological spoilage and the oxidative deterioration of refrigerated foods. For instance, Rodrigues et al. [3] incorporated a natural extract from green tea and ginger into a corn starch-based packaging to preserve the freshness and shelf life of strawberries. In another study, the antimicrobial and antioxidant effect of active packaging enriched with corn stigma residue extracts on the conservation of refrigerated meat was investigated. The presence of bioactive compounds slowed the increase in pH, strictly related to microbial enzymatic activities, and reduced the lipidic oxidation by up to 60% [1]. Moreover, Ortenzi et al. investigated the positive effect of PLA samples functionalized with natural phenolic compounds on the extension of the refrigerated salami shelf life [19].

Bioactive extracts from spent coffee grounds (SCG) can be exploited to act as antioxidants in food active packaging due to their large amount of caffeine, phenolic compounds, flavonoids, chlorogenic acids, and melanoidins [20,21,22]. Many extraction processes, both conventional and not, have been investigated in order to obtain a high yield in terms of the antioxidant power of the extract [23,24,25,26], which can be then incorporated into the polymer matrices. Among them, bio-based and biodegradable polymers, such as polylactic acid (PLA), represent a promising solution to disposal, environmental, and health issues raised from oil-based plastics [27,28]. PLA is a thermoplastic polymer, synthesized from the lactic acid monomer, that is mainly produced by the bacterial fermentation of starch-rich products, such as corn, sugar beet, tapioca, and wheat [29,30]. It is soluble in organic solvents, including dioxane, acetonitrile, chloroform, methylene chloride, 1,1,2-trichloroethane, and dichloroacetic acid [30]. PLA shows numerous advantages, including biocompatibility, safety for food applications, excellent transparency, good water resistance, and good processability with standard plastic techniques. Among them, a possibility is represented by solvent casting, which, on the other hand, entails the eventual use of hazardous solvents, such as chloroform. Although its degradation process leads to CO_2_ formation, PLA emits less greenhouse gases compared to the oil-based polymers. In fact, if analyzed from the cradle to the factory, CO_2_ residual emissions are negative because the carbon dioxide emitted by the process of degradation is the same as the natural feedstock to lactic acid production absorbed from the air during its growth [31]. Nevertheless, it is not as widespread as conventional polymers at the moment because of its poorer mechanical characteristics and gas barrier properties [30,31,32,33]. In order to improve its mechanical properties, nanoparticles are usually added as fillers, such as lignocellulosic materials, while the gas barrier properties can be enhanced through the incorporation of some additives, including SCG extracts [34]. The beneficial effects of SCG extracts on active packaging captured the interest of several authors, working on different biopolymers, including pectin, zein, and polylactic acid. Nevertheless, the extract properties and the provided features on the final active packaging strongly change as a function of the operating conditions of the extraction method, extract purification, characteristics, and in combination with the selected polymer and production technique [34,35,36,37,38,39].

Therefore, in this work the optimization of the production process of an innovative active food packaging was carried out. In particular, a PLA-based matrix was enriched by different concentrations of different SCG extracts, proposing a sustainable alternative to conventional packaging due to the use of a biodegradable polymer and the valorization of waste. Moreover, the characterization of the resulting packaging was provided, and its suitability for refrigerated foods was tested.

## 2. Materials and Methods

### 2.1. Chemicals and Spent Coffee Grounds Preparation

Ethanol, methanol, acetonitrile, glacial acetic acid, chloroform, dichloromethane, PEG 400, 2,2′-azino-bis (3-ethylbenzothiazoline-6-sulphonic acid) diammonium salt, potassium persulfate, sodium thiosulphate, potassium iodide (Carlo Erba, Cornaredo, Italy), starch, Folin–Ciocalteu’s reagent, sodium carbonate and reference standards (caffeine and chlorogenic acid) (Sigma Aldrich, Milan, Italy), and polylactic acid (PLA, Ingeo™ Biopolymer 2003D, NatureWorks Italia S.r.l., Milan, Italy) were of analytical grade, and they were used without further purification. Deionized water (Idrotecnica, Genova, Italy) and ultrapure water (Milli-Q System, Millipore Inc., Burlington, MA, USA) were used.

The spent coffee grounds (*Coffee canephora* variety) were collected from an espresso vending-machine at the Department of Civil, Chemical, and Environmental Engineering of the University of Genoa. They were dried at 45 °C until approximately 6% of the residual moisture was reached. The dried product was stored at room temperature, under dark conditions and inside water-proof containers.

### 2.2. PLA-Based Film Preparation by Solvent Casting

The PLA-based solutions were prepared by dissolving PLA in chloroform under magnetic stirring (500 rpm) at room temperature for 6 h. After the polymer’s complete dissolution, the obtained solutions were sonicated in an ultrasonic bath (700 W, 1 min) to remove air bubbles, and 5 mL of the product was poured on 9-cm diameter glass Petri plates. Chloroform removal was performed under the hood until complete evaporation before peeling the films from the mold. A first screening of the variables involved in the film production was undertaken, analyzing the effect of the polymer-to-solvent ratio (4%, 5%, 6%, 8% *w*/*v*), solvent drying time, and the concentration of the plasticizer PEG 400 (0%, 10%, 20% *w*/*w*) with respect to the polymer mass on the macroscopic features of the film (homogeneity, transparency, thickness/handling).

### 2.3. Extracts Preparation

The antioxidant compounds to be used as active agents for the PLA-based food packaging were obtained by SCG by adopting three different approaches.

The first extract was produced by high-pressure high-temperature extraction (HPTE), which was performed by a lab-scale stainless-steel stirred discontinuous extractor (Parr Instruments Company, model 350M—4650 Series, Moline, IL, USA) [25]. The extraction was carried out at 150 °C and 7.2 bar for 1 h, with a liquid-to-solid ratio of 10 mL/g, under inert atmosphere, and using ethanol 54% *v*/*v* as solvent, in agreement with the previous optimization studies [40]. After the extraction, the solid residue and liquid phase were separated by filtration (1.2 μm) and stored at −20 °C.

A second potential active agent to be tested was obtained by the purification of the HPTE extract by liquid-liquid extraction (LLE) with chloroform, due to the higher solubility of caffeine in the non-polar phase [41] and chloroform being the same solvent used for PLA-based film production. In a 50 mL flask, the two phases were mixed at room temperature using a liquid-to-liquid ratio of 0.5 mL _extract_/mL _chloroform_. The mass transfer from one phase to the other was obtained by gentle mixing, and the successive natural separation of the two phases took place in approximately 1 h. The non-polar phase was then recovered and stored at −20 °C.

A third extract was obtained by solid–liquid extraction (SLE) from dried SCG using chloroform as solvent. The process was carried out in a glass lab-scale discontinuous extractor, equipped with a condenser, and heated by an electrical jacket. The extraction was carried out by setting a temperature of 70 °C, at 1 atm, for 2 h [34], using a liquid-to-solid ratio of 10 mL/g. After the extraction, the solid residue and liquid phase were separated by filtration (1.2 μm), and the extract was stored at −20 °C.

### 2.4. Extracts Characterization and Comparison

The extracts produced by HPTE, LLE, and SLE were analyzed in terms of extract total solids, antiradical power, total polyphenol, caffeine, and chlorogenic acid content.

The extract total solids were determined by gravimetric analysis. First, 2 mL of the extract was dried at 110 °C until a constant weight was reached. The extracts obtained by LLE and SLE, containing chloroform, were previously subjected to a complete evaporation of the solvent under the hood at room temperature. The total solids were expressed as milligrams of total solids per gram of dried SCG.

The antiradical power (ARP) of the extracts was evaluated by the ABTS^•+^ assay [42], expressed as micrograms of Trolox equivalents (TE) per gram of dried SCG.

The chloroform-based extracts were pre-treated to be suitable for the analyses of ARP, total polyphenols (TP), and caffeine and chlorogenic acid content. Complete solvent disposal was performed in rotavapor at 40 °C under vacuum (model Laborota 4000, Heidolph Instruments, Schwabach, Germany), and the samples were redissolved in ethanol and analyzed according to the protocol reported by [25]. TP expressed as milligrams of caffeic acid equivalent (CAE) per gram of dried SCG, was determined by a modified Folin–Ciocalteu’s assay [25].

The caffeine and chlorogenic acid content in the extracts, expressed as milligrams per gram of dried SCG, were investigated by using high-performance liquid chromatography (HPLC, Agilent, 1100 Series, Palo Alto, CA, USA) equipped with a C18 phase column (Model 201TP54, Vydac, Hesperia, CA, USA), combined with a diode-array detector (DAD), following the method described by Aliakbarian et al. [43].

### 2.5. Lipid Peroxidation Inhibition of HPTE Extract

The effect of the HPTE extract on the inhibition of the lipid peroxidation in stressed extra virgin olive oil (EVO) was evaluated. To avoid the direct mixing of HPTE extract with the oil, the antioxidant-rich extract was previously encapsulated into maltodextrins (DE 16.5–19.5, Sigma Aldrich, Milan, Italy) by spray drying (Mini Spray Dryer B-290, BÜCHI Labortechnik AG, Flawil, Switzerland). Then, 10 g of polymer were poured into 100 mL of the hydroalcoholic extract and left under stirring until the complete dissolution of maltodextrins at room temperature. The solution was then fed (feed flow rate = 7.5 mL/min) to the spray dryer working at 160 °C as inlet temperature and with an aspiration rate of 32 m^3^/h. The obtained particles were analyzed in terms of moisture, product recovery, and encapsulation efficiency, following the methods described by Pettinato et al. [26].

Lipid peroxidation inhibition tests were performed by exposing EVO and EVO in which encapsulated extract (10% *w*/*v*) was added in stressing conditions. Lipid peroxidation was induced by bubbling pure oxygen for 2 min into 20 mL of the sample and exposing hermetically sealed tubes containing the oxygen-saturated samples to the sunlight for 3 days. Furthermore, samples of EVO with and without maltodextrins only were stored in the dark and oxygen free during the experiments as reference samples. The peroxide value (PV) was evaluated by titration, following the protocol reported in the Annex III of the Commission Regulation (EEC) No 2568/91 of 11 July 1991 [44]. The peroxide value, expressed as milliequivalents (meq) of active oxygen per kilogram, was calculated as in Equation (1):(1)$$\mathrm{PV}\;\left(\frac{{\mathrm{meq}\;}_{\mathrm{active}\;\mathrm{oxygen}}}{g}\right)=\frac{V\cdot T\cdot 1000}{m}$$
where *V* is the volume (mL) of the standardized sodium thiosulfate solution used for the analysis; *T*, expressed as mol/L, is the molarity of the sodium thiosulfate solution (0.01 mol/L); and *m* is the mass of the EVO sample.

### 2.6. PLA-Based Film Loaded with SCG Extracts

The samples of active packaging were prepared by dissolving PLA in chloroform (8% *w*/*v* polymer-to-solvent ratio) under stirring (500 rpm) at room temperature. Various aliquots of the previously obtained extracts were added to the solutions. Due to the different composition of the obtained extracts, the theoretical loading of the active agent was expressed in terms of the mass of loaded caffeine per mass of polymer (µg _caffeine_/g _PLA_). For the theoretical loading definition, and to make uniform the procedure adopted in further experiments (i.e., migration tests), the caffeine content was determined via UV-vis spectrophotometer (model Lambda 25, Perkin-Elmer, Wellesley, MA, USA). With this purpose, the caffeine in the extract was separated by liquid–liquid extraction using dichloromethane [45]. Then, 1 mL of the sample and 5 mL of dichloromethane were mixed for a few minutes, then were placed in a separating funnel where the caffeine was extracted. The process was repeated 3 times, and the solvent layers combined, and the absorbance was read at 260 nm into quartz cuvettes. The caffeine concentration (C, mg/mL) was calculated from sample absorbance (ABS _260 nm_) through a calibration curve (Equation (2)) obtained from standard solutions of caffeine in dichloromethane.
(2)$${\mathrm{ABS}}_{\;260\;\mathrm{nm}}=28.2\;\cdot \;C\left(R^{2}=0.9996\right)$$

Particularly, 512, 676, and 840 µg _caffeine_/g _PLA_ of the HPTE extracts and 512, 676, 840, 1167, and 1494 µg _caffeine_/g _PLA_ of the LLE extracts were tested. The extracts and polymer solution (Figure 1) were mixed for 6 h, then sonicated in an ultrasonic bath (700 W, 1 min) to remove the air bubbles. Finally, 5 mL of the product was poured on 9-cm diameter glass Petri plates. The chloroform removal was performed under the hood until complete evaporation before peeling the films from the mold.

### 2.7. Packaging Characterization

#### 2.7.1. Morphology and Thickness

Images of the casted films and their thicknesses were obtained by an Olympus BX51 transmitted and reflected light microscope.

The surface morphology of the samples was investigated by an ultra-high resolution field-emission-source scanning electron microscope (SEM, Zeiss CrossBeam XB 1540, Carl Zeiss, Milan, Italy) at an accelerating voltage of 5 kV and 2 kV and at a working distance of approximately 3 mm. The films were first coated with a thin layer of chrome by a high vacuum sputter coater (Emitech K575X, Lewes, UK) to avoid charging effects.

#### 2.7.2. Release Tests

The ability of active packaging samples to deliver compounds exhibiting antioxidant activity in the environment surrounding food was evaluated by release tests. Films (2 × 2 cm^2^) enriched with SCG extracts at different concentrations were placed in contact with 6 mL of 10% ethanol (*v*/*v*) as a food simulant for 48 h under mild agitation. The tests were carried out at room temperature to first evaluate a stricter storage condition than the refrigerated one. The food simulant was chosen according to the European Regulation on plastic materials and objects intended to come into contact with food [46,47].

The protocol reported by da Silva et al. [25] was used to perform the ABTS^•+^ assay, which was carried out on the food simulant sampled at different times in order to assess the antiradical power of the released compounds.

#### 2.7.3. Gas Permeability and Diffusivity Measurements

The films’ gas barrier properties towards oxygen and carbon dioxide were evaluated by using an innovative ultra-high-vacuum apparatus based on membrane techniques, proposed by Firpo et al. [48]. It allows measures of the permeability, diffusivity, and solubility of gases with a molecular mass lower than 100 amu. In this way, materials with very different transport properties can be characterized. The method reported by Firpo et al. [48] was applied to perform the tests.

The packaging sample, with a thickness of approximately 40 μm and an exposed area of 7.1 mm^2^, was mounted on a perforated copper disk compatible with ultra-high-vacuum flanges CF16, then located between an upstream and a downstream high-vacuum chamber. A thin layer of Torr Seal^®^ was then spread on the interface between the copper disk and the sample, ensuring vacuum sealing and negligible leakage. A schematic representation of the assembly is shown in Figure 2. Before any measurements, the assembly was tested by checking the mechanical seal at 1 atm of upstream pressure of air. Furthermore, a leak test carried out with helium verified the seal of the flanges by monitoring the signal in the downstream chamber with a residual gas analyzer (RGA).

The apparatus can work both at constant pressure (dynamic method) and at constant-volume–variable-pressure (static method), in agreement with the standard test method [49].

Regarding the diffusivity measurements, they were carried out in dynamic mode. In this case, the experimental results were recorded in terms of ion currents by the RGA. The gas diffusivity was then determined by the lag time method. Assuming that the gas was introduced at *t* = 0 from the upstream side of the sample, the amount of gas *Q*(*t*) that passed through the membrane at time *t* > *t** can be expressed as reported in Equation (3), where *t** is the time at which the steady-state condition was reached.
(3)$$Q\left(t\right)=A(\int_{0}^{t^{*}}J\left(t\right)dt+\int_{t^{*}}^{\tau }J_{ss}dt)$$
where *J*(*t*) is the rate of gas transfer per unit of area, *J_ss_* is the rate at the steady-state, and *τ* is the total measurement time.

Comparing this equation with the Equation (4.24) reported by Crank [50], and considering the proportionality between the rate of transfer of the tracer gas per unit of area of the membrane and the ionic current measured by the RGA, it was possible to obtain the diffusivity, as shown in Equation (4).
(4)$$D=\frac{L^{2}\;\left(I_{ss}-I_{0}\right)}{6\int_{0}^{t^{*}}I_{ss}-I\left(t\right)dt}$$
where *I_ss_* is the ion current of the tracer gas at the end of the measure, when it was expected to represent a good approximation of the steady-state value; and *I*_0_ is the background ion current of the tracer gas, just before the input of the gas in the upstream chamber.

The integral in Equation (4) was calculated numerically through the trapezoidal rule by considering the data points recorded by the RGA at a sampling time of 2 s.

Since the RGA sensitivity could be estimated with a not negligible uncertainty, given by the calibration of the instrument, the permeability was determined through the static method with more accuracy [48]. In this case, the rate of the downstream pressure *p_d_* rise over time was monitored by a spinning rotor gauge (SRG) until the collected results were enough for the post-processing. The background due to degassing was subtracted from the overall curve and a fit with the diffusion equation (Equation (4.24), [50]) by the least squares method was performed. The slope of the fitted line ∂*p_d_*/∂*t* was then used to calculate the steady state flux *J =* (*V*/*A*) ∂*p_d_*/∂*t*, where *V* is the vacuum chamber volume and, consequently, the permeability coefficient *P* by the following equation [48]:(5)$$P=\frac{JL}{\Delta p}$$

Finally, the solubility *S* of the tracer gas was calculated according to Henry’s law (Equation (6)):(6)$$S=\frac{P}{D}$$

The relative uncertainties of the measurement setup were also considered. The permeability, the diffusivity, and the solubility presented a relative uncertainty of 13%, 9%, and 22%, respectively.

#### 2.7.4. Migration Tests

Migration tests at 4 °C were carried out on the PLA-based films enriched with SCG extracts and in 10% ethanol (*v*/*v*) as food simulant [43,44,48]. The films (2 × 2 cm^2^) were put into contact with 20 mL of food simulant for 8 days under static conditions, and at the end of the tests, the overall migration and specific migrations of caffeine were evaluated. The specific migration of caffeine was determined via UV-vis spectrophotometer (model Lambda 25, Perkin-Elmer, Wellesley, MA, USA), using the method proposed by Belay et al. [45] with some modifications. Briefly, a liquid–liquid extraction in 2 mL of dichloromethane was performed on samples of the food simulant. Depending on the caffeine content, a variable volume of the food simulant samples (from 1 to 6 mL) was used to extract a detectable amount of caffeine. After three hours of contact time, the extractant phase was manually withdrawn, and its absorbance was measured into quartz cuvettes at a wavelength of 260 nm. The caffeine concentration (C, mg/mL) was calculated from sample absorbance (ABS _260 nm_) through the calibration curve reported in Equation (2).

#### 2.7.5. Lipid Peroxidation Inhibition Ability

The ability of films loaded with an active agent in preventing lipid oxidation was evaluated. First, 20 mL of EVO was poured into a 50 mL tube, whose lateral surface in contact with the sample was internally coated with 42.4 cm^2^ of the films enriched with different concentrations of the extracts. The samples were then exposed first to the UV light for three days and to sunlight and oxygen for a further three days, at room temperature. The peroxide values of the samples in contact with the active packaging and with reference samples (EVO), exposed to the same stressing conditions, were determined by titration [44].

#### 2.7.6. Statistical Analysis

The statistical analysis was carried out using Statistica software 6.0 (StatSoft, Tulsa, OK, USA). One-way analysis of variance (ANOVA) and Tukey’s post hoc test were performed to determine the statistically significant differences among the samples.

## 3. Results and Discussion

### 3.1. Optimization of Solvent Casting Technique for PLA Film Production

PLA-based films were produced by using the solvent casting method. First, an optimization study was undertaken to investigate the effect of the polymer-to-solvent ratio and plasticizer concentration (PEG 400) on the features of the obtained films. PEG 400 concentrations from 0 to 20% (*w*/*w*) were tested, but the presence of the plasticizer led to low polymer transparency and halos, as shown in Figure 3a, without significant advantages in terms of versability of the polymer solution, homogeneity of the film, and surface smoothness. Concerning the polymer-to-solvent ratio, this variable affected the solution versability being directly related to the solution viscosity, i.e., polymer-to-solvent ratios below 8% *w*/*v* provided films that were too thin, hard to be peeled from the mold, and fragile. It did not significantly affect the final film thickness, which comprised results between 40 and 50 µm at the conditions tested. Furthermore, since chloroform is extremely volatile, the initial solvent evaporation rate was decreased by covering the molds with a perforated aluminum foil, and a good transparency was obtained. An optimal drying time of 72 h was finally selected. Figure 3 shows some films obtained in the various process optimization steps. Therefore, the optimal conditions to produce the films were a polymer-to-solvent ratio of 8% *w*/*v*, without the addition of plasticizer, and a drying time of 72 h.

### 3.2. Extract Characterization and Comparison

Table 1 shows the experimental results derived from the characterization of the extracts obtained by the three different approaches from the same dried biomass (SCG). The data obtained for the hydroalcoholic extract are in agreement with those reported in a previous study [40] with slight differences due to the variability of the used biomass. The extract obtained with HPTE exhibited the highest amount of total solids, followed by those derived from SLE and LLE, in which the solvent better promoted the recovery of lipidic compounds and caffeine than polyphenols and chlorogenic acid. SLE provided the lowest content of caffeine compared to HPTE and LLE, showing how the effects of the high temperature and pressure in HPTE can improve the mass transfer, providing high extraction yields, even using less selective (but greener) solvents. The purification of the HPTE extract with LLE was performed to enable the use of a larger volume of extract (and caffeine) inside the active packaging due to the higher compatibility between the polymer solution and the purified extract. The use of chloroform in LLE depleted the composition of the HPTE extract; indeed, after the purification, polyphenols and chlorogenic acid were not detected in the extracting phase, and the caffeine yield was also reduced (Figure 4b). Further evidence of this effect was provided by the analysis of the polar phase of the LLE by Folin–Ciocalteu’s assay. It presented the same polyphenol content of the hydroalcoholic extract, within the experimental error (36.2 ± 1.1 mg _CAE_/g _SCG_), highlighting the low affinity of polyphenolic compounds, including chlorogenic acid, toward chloroform [51]. Consequently, a decrease in antioxidant activity was observed from HPTE to LLE extract. SLE provided the product with the lowest antioxidant activity due to the negligible presence of polyphenols and the assay used for the detection of antioxidant activity. Indeed, caffeine is not able to scavenge ABTS^•+^ radicals, but it is effective as a hydroxyl radical scavenger [52]. Nevertheless, the caffeine content in the LLE extract was significantly higher than in the extract produced via SLE (Figure 4b,c). Therefore, the hydroalcoholic extract and the one obtained by its purification through LLE showed better performances if compared to the direct solid–liquid extraction in chloroform. The last was considered unsuitable for the successive active food packaging applications.

### 3.3. Lipid Peroxidation Inhibition of HPTE Extract

The extract provided by HPTE was encapsulated into maltodextrins by spray drying before testing its ability to prevent the lipid peroxidation of EVO under stressing conditions. The spray drying process supplied a solid and dry product with an 80 ± 0.05% product recovery and a residual moisture of approximately 0.114 ± 0.002 g _water_/g _wet powder_. The encapsulation efficiency was determined basing on the total polyphenol content of the fed extract and the dry product [53]. The produced particles showed a total polyphenol content of 18.2 ± 0.7 mg _CAE_/g _dried solids,_ of which 4.5 ± 0.3 mg _CAE_/g _dried solids_ were superficial polyphenols (not encapsulated). Therefore, an encapsulation efficiency of 61.5 ± 3.5% was obtained. The effect of the spray-dried extract on the prevention of EVO lipid peroxidation was then evaluated by adding 10% (*w*/*v*) dry extract into 20 mL of EVO and exposing the samples to pure oxygen and sunlight as stress factors to induce lipid peroxidation. Figure 5a shows the data related to control samples, which were used to assess potential interferences in the PV evaluation. The EVO control sample, stored in the dark and under an oxygen-free environment, exhibited a constant PV over time. The statistical analysis on the results of the control samples demonstrated that there were no significant differences (*p* > 0.05) between the values of PV of the samples with only EVO and with encapsulated extract (*p* = 0.0551), while a PV slightly higher was shown by the sample in which pure maltodextrins were added (*p* = 0.0036). The first result implies that the presence of encapsulated extract did not interfere with the titration method for PV determination, while the presence of only maltodextrins determined a little interference, but conservative, in the PV determination. In Figure 5b, the PV of samples exposed for 3 days to sunlight and saturated with pure oxygen are shown. EVO without the encapsulated extract exhibited a PV 10 times higher than EVO stored for 3 days in the dark and oxygen free (control sample), while the presence of the encapsulated extract allowed a reduction in lipid peroxidation (71% lower).

These results agree with the literature. Indeed, Naz et al. [54] studied the oxidative stability of several types of oils under oxidative deterioration conditions, and the exposure to air and light resulted as the most effective stress condition in increasing the PV of the sample. Furthermore, the addition of antioxidants to the tested samples (caffeic, vanillic, and ferulic acids) provided a reduction in the PV, and particularly, caffeic acid, which is contained in the structure of chlorogenic acid, resulted as the most active in preventing lipid oxidation. Moreover, Hwang et al. [55] investigated the effect of SCG extracts on the inhibition of the oxidation of soybean and fish oils, evidencing their great potential as natural antioxidants.

### 3.4. Characterization of Packaging Enriched with SCG Extracts

Optimized protocols for PLA-based film production were adopted to fabricate active packaging. The SCG extracts were loaded into the polymer at different concentrations, expressed as caffeine theoretical loadings, as follows: 512, 676, and 840 µg _caffeine_/g _PLA_ for HPTE extracts and 512, 676, 840, 1167, and 1494 µg _caffeine_/g _PLA_ for LLE extracts. Due to the low quantity of bioactive compounds detected in the SLE extract, it was not selected for further steps of the study. The films enriched with the extract obtained by HPTE showed a more intense color as the extract concentration increased. The color change was probably related to an increase in polyphenolic compounds and melanoidins (Figure 6b). The extract was added until the film homogeneity was still acceptable, i.e., up to a theoretical loading of 840 µg _caffeine_/g _PLA_, but the best results were obtained with a theoretical loading of 512 µg _caffeine_/g _PLA_. Indeed, due to the different solvents used for the extraction and polymer dissolution, higher loadings of HPTE extract provided a product with localized extract aggregates that were non-transparent and had high heterogeneity. The films enriched with the hydroalcoholic extract exhibited a higher handling than those with the extract obtained by LLE, but the film transparency was negatively affected, even at lower concentrations. The films enriched with the extract from LLE, on the contrary, were less brown-colored (Figure 6c) and more transparent, also at a high theoretical loading. Therefore, in the last case, it was possible to incorporate a larger amount of the extract, up to 1494 μg _caffeine_/g _PLA_, improving the antioxidant potential of the packaging.

#### 3.4.1. Release Tests

Release tests were performed in 10% ethanol (*v*/*v*) as food simulant, at room temperature, and under stirring for 48 h. Table 2 shows the antiradical power associated with the release of antioxidant compounds from the packaging to the food simulant over time. Samples with the same theoretical loadings were employed for comparison purposes.

All the films showed a release of antioxidant compounds that was quite constant within the 48 h of tests. The obtained data are the results of a balance between the releasing rate and the degradation reactions of the released compounds due to the exposure to oxygen and light. However, the constant ARP observed indicates that releasing and degradation rates are comparable, allowing a constant antioxidant action of the active agent towards food. The same behavior was observed in a previous work for zein-based packaging obtained by solvent casting and using SCG extract as an active agent [38]. Particularly, food simulant samples in contact with films enriched with the HPTE extract exhibited higher values of ARP than those in contact with the extract obtained by LLE. This evidence can be ascribed to the higher antioxidant content of the HPTE extract and polyphenols higher affinity towards the simulant used. Nevertheless, non-significant dependencies were observed on the theoretical loading for both the active agents employed. For this reason, and considering the better macroscopic features, films enriched with the lowest HPTE extract loadings were considered the best solution and used for further tests. Concerning the active packaging loaded with the LLE extract, the highest extract loading was evaluated in further tests.

#### 3.4.2. Morphology and Thickness

The morphology of the three representative films obtained by the solvent casting technique was examined by both optical microscope and SEM. Figure 7 shows images taken for the blank sample (Figure 7a) and those enriched with the hydroalcoholic extract (Figure 7b) and the extract further purified (Figure 7c), with caffeine contents of 512 μg _caffeine_/g _PLA_ and 1494 μg _caffeine_/g _PLA_, respectively.

The reference sample showed more surface irregularities than those enriched with the extracts, maybe due to a heterogeneous aggregation of the polymer particles during the drying process, as attested by Bisharat et al. [56]. Moreover, some defects could be due to the formation of air bubbles and the film damage during its removal from the mold. The addition of the hydroalcoholic extract led to a more continuous surface and a compact structure, probably as a result of a better polymer matrix filling, as also attested by Mendes et al. [35] for pectin films enriched with SCG extracts. In addition, the plasticizing effects on PLA due to SCG extracts were observed in different studies [36,37,39]. On the other hand, it exhibited the presence of dark beads of heterogenous dimensions that could be due to the poor extract miscibility in chloroform. Finally, the extract obtained by LLE showed a more porous morphology, more discernible at SEM.

Moreover, the films’ thickness, measured through the optical microscope, ranged between 40 and 50 μm. Variability in the values were mainly affected by the production process rather than by the presence of the extracts.

#### 3.4.3. Gas Permeability and Diffusivity Measurements

CO_2_ and O_2_ permeability and diffusivity measurements were carried out on three different representative samples: the PLA-based film without extract loading (reference sample), films enriched with the HPTE extract (theoretical loading of 512 μg _caffeine_/g _PLA_), and the extract further purified via LLE (theoretical loading of 1494 μg _caffeine_/g _PLA_).

Figure 8 gives a comparison between O_2_ and CO_2_ permeabilities, diffusivities, and solubilities of the three samples. The samples composed of PLA only showed values of permeabilities and diffusivities comparable with those in the literature, both for O_2_ and CO_2_ [57], while O_2_ permeability decreased when the films were enriched with the SCG extracts. This behavior was correlated to the decrease in diffusion, probably due to the antioxidant action of the extracts. Instead, both of the SCG extracts did not significantly affect O_2_ solubility, since it ranged between 3.48·10^−2^ and 3.99·10^−2^, values comparable within the relative uncertainties. Regarding CO_2_ permeability, it slightly decreased when films were enriched with the coffee extracts. This trend was mainly affected by the decrease in CO_2_ solubility, since the presence of the extracts probably affected CO_2_ condensability, strictly related to its solubility [58], as well as any type of gas–polymer interactions and the polymer morphology. Instead, CO_2_ diffusivity increased as the extract concentration increased. This was presumably ascribed to the fact that both the lipidic compounds contained in the extracts and CO_2_ had a plasticizing effect on the polymer matrix, leading to a higher chain mobility and probably to a larger free volume. However, if compared to CO_2_, O_2_ showed the lowest values in permeability but the highest in diffusivity, since the latter mainly depends on the gas size: O_2_ is smaller than CO_2_ and it diffuses better. However, since CO_2_ condensability is higher than that of O_2_, CO_2_ presented the highest values of solubility.

#### 3.4.4. Migration Tests

To evaluate the behavior of the packaging at longer times and under refrigerated conditions, migration tests were carried out at 4 °C for eight days in 10% ethanol (*v*/*v*) as food simulant. According to a first qualitative investigation, all the sample types maintained their transparency, and they showed a negligible loss in weight, below 0.35%.

The specific migration of caffeine was evaluated via UV-vis spectrophotometer after treating the food simulant samples with a further liquid–liquid extraction in dichloromethane. The obtained results are reported in Figure 9.

The reference sample presented a not negligible value in terms of caffeine content. This was probably due to an interference recorded at the same wavelength of the caffeine peak, as demonstrated by an UV-Vis spectrophotometric analysis. In fact, it showed a caffeine migration peak between 230 nm and 275 nm, and a peak at 274 nm was also recorded for the caffeine standard in dichloromethane at a concentration of 10 mg/L.

Regarding the other samples, the caffeine migration as a function of theoretical loading showed not statistically significant differences among films loaded with HPTE extract_,_ while the wider range of theoretical loading allowed by LLE extract enabled us to observe a significant dependance of migrated caffeine on the active agent loaded in the polymer. Nevertheless, the films loaded with the extract obtained by HPTE showed caffeine migration values close to their initial theoretical loading, as opposed to those containing LLE extracts, which presented a slower caffeine migration. This was probably due to the greater compatibility between the extract and the solvent used for casting, given the higher solubility of caffeine in the non-polar phase. Indeed, a better polymer matrix filling and a more compact final structure were obtained, as demonstrated by the studies on the film morphology.

#### 3.4.5. Lipid Peroxidation Inhibition Ability

The antioxidant action of the films enriched with the extracts was also evaluated in terms of ability to inhibit EVO lipid peroxidation. First, the EVO samples were stressed by exposure to UV light for three days at room temperature. The peroxide value (PV) trend in the tested samples is reported in Figure 10.

As expected, all the samples under stressing conditions showed a significantly higher PV compared to the control sample, which was stored in the dark and under an oxygen-free environment. The only exception was given by the EVO in contact with the extract obtained by HPTE at the theoretical loading of 840 µg _caffeine_/g _PLA_, whose PV resulted as not significantly different (*p* > 0.05) from the control. In general, the samples loaded with HPTE extract exhibited peroxidation inhibition performances higher than samples with LLE extract at the same loadings. By prolonging the exposure of samples and changing UV with sunlight (Figure 11), an intensification of the protecting effects of the extracts could be observed. Whereas the PV of the control sample was not affected by duplicating the storage time, a remarkable increase could be noticed for the blank (from 12.9 ± 0.9 to 20.7 ± 3.3 meq _active O2_/kg _EVO_) and reference samples (from 12.4 ± 0.0 to 19.5 ± 0.2 meq _active O2_/kg _EVO_). The results showed that theoretical loadings of both extracts higher than 676 µg _caffeine_/g _PLA_ are able to provide an inhibition of lipid peroxidation, and in particular, active films loaded with the extract obtained by HPTE. However, for films enriched with LLE extracts, there can be observed a growing effect by increasing the extract loading, while HPTE extract provided a stronger intensity in inhibiting peroxidation but without a clear trend as a function of the theoretical loading. The hydroalcoholic extract, due to the presence of a complex pool of antioxidants and the browning effect on the film, is expected to exert a higher inhibition compared to the LLE extract at the same value of theoretical loading, but the highest loading capability of films produced using LLE extract allows us to reach comparable effects to HPTE extract by increasing the loading values.

## 4. Conclusions

This study investigated the production of an innovative active food packaging with an intended use for food stored under refrigerated conditions. Bioactive compounds were recovered from SCG through HPTE, and the extract exhibited significant antiradical power (0.57 ± 0.06 μg TE/g SCG), polyphenol (36.4 ± 1 mg CAE/g SCG), caffeine, and chlorogenic acid contents (10.3 ± 0.18 mg caffeine/g SCG and 2.4 ± 0.06 mg chlorogenic acid/g SCG). Moreover, an alternative active agent was obtained via purifying the HPTE extract with LLE using chloroform. Process optimization enabled the possibility of producing active packaging based on SCG extracts as antioxidant agents. The obtained packaging films presented a good transparency and handling, with a thickness ranging between 40 and 50 μm. The two different extracts provided different effects on the final structure and active action performances, i.e., it reduced the surface irregularities, but the extract obtained by HPTE caused the presence of dark beads on the films’ surfaces, while the film enriched with LLE extract showed a slightly less regular surface. The presence of the active agent enhanced the gas barrier properties of the polymer, probably due to the antioxidant action and to a plasticizing effect.

Migration tests indicated that films loaded with the LLE extract provided a lower caffeine migration rate and, in general, a lower significant action than packaging loaded with HPTE extracts. Therefore, the latter showed the best performances in terms of lipid peroxidation inhibition ability and antiradical power due to the presence of a complex pool of antioxidants and the browning effect on the film, despite the limited loading capacity (840 μg caffeine/g PLA) compared to the LLE extract.

Among the future perspectives of this research work, an investigation of the potential impact of the obtained films on the food sensory properties, such as taste and texture, is planned. Indeed, it could be a valuable insight to individuate the most suitable applications of the produced active packaging.

## Figures and Tables

**Figure 1 foods-12-04167-f001:**
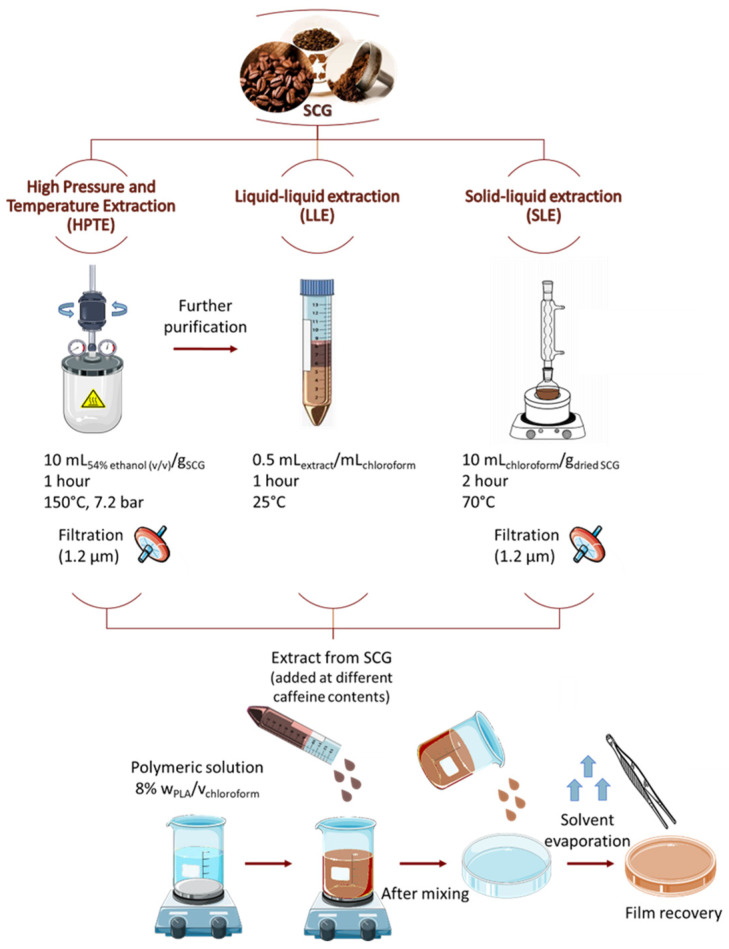
Schematic representation of active packaging production.

**Figure 2 foods-12-04167-f002:**
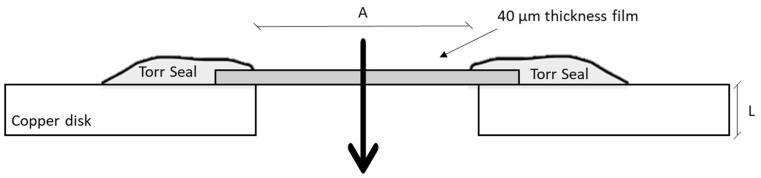
Schematic representation of the membrane assembly. Adapted from Firpo et al. [48].

**Figure 3 foods-12-04167-f003:**
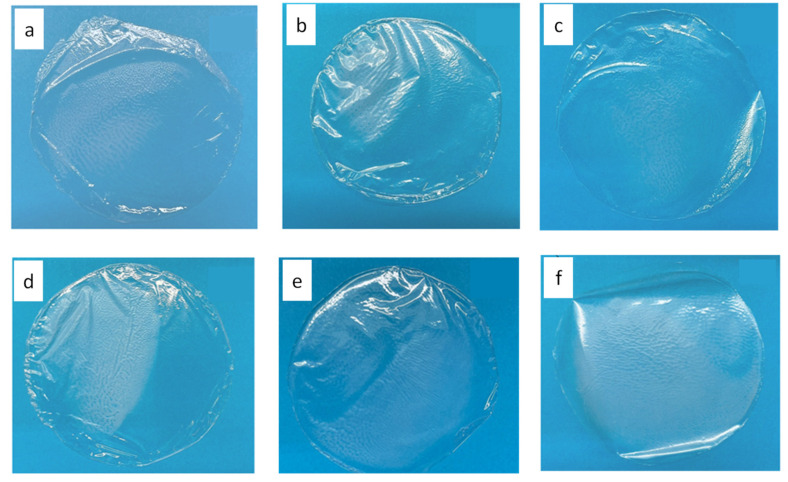
PLA-based films obtained using (**a**) polymer-to-solvent ratio = 4% (*w*/*v*), PEG 400 = 0% (*w*/*w*), drying time = 12 h; (**b**) polymer-to-solvent ratio = 4% (*w*/*v*), PEG 400 = 10% (*w*/*w*), drying time = 12 h; (**c**) polymer-to-solvent ratio = 4% (*w*/*v*), PEG 400 = 20% (*w*/*w*), drying time = 12 h; (**d**) polymer-to-solvent ratio = 5% (*w*/*v*), PEG 400 = 0% (*w*/*w*), drying time = 12 h; (**e**) polymer-to-solvent ratio = 5% (*w*/*v*), PEG 400 = 20% (*w*/*w*), drying time = 12 h; (**f**) polymer-to-solvent ratio = 5% (*w*/*v*), PEG 400 = 20% (*w*/*w*), drying time = 12 h; (**g**) polymer-to-solvent ratio = 6% (*w*/*v*), PEG 400 = 0% (*w*/*w*), drying time = 12 h; (**h**) polymer-to-solvent ratio = 6% (*w*/*v*), PEG 400 = 10% (*w*/*w*), drying time = 12 h; (**i**) polymer-to-solvent ratio = 6% (*w*/*v*), PEG 400 = 20% (*w*/*w*), drying time = 12 h; (**j**) polymer-to-solvent ratio = % (*w*/*v*), PEG 400 = 0% (*w*/*w*), drying time = 12 h; (**k**) polymer-to-solvent ratio = 6% (*w*/*v*), PEG 400 = 0% (*w*/*w*), drying time = 72 h.

**Figure 4 foods-12-04167-f004:**
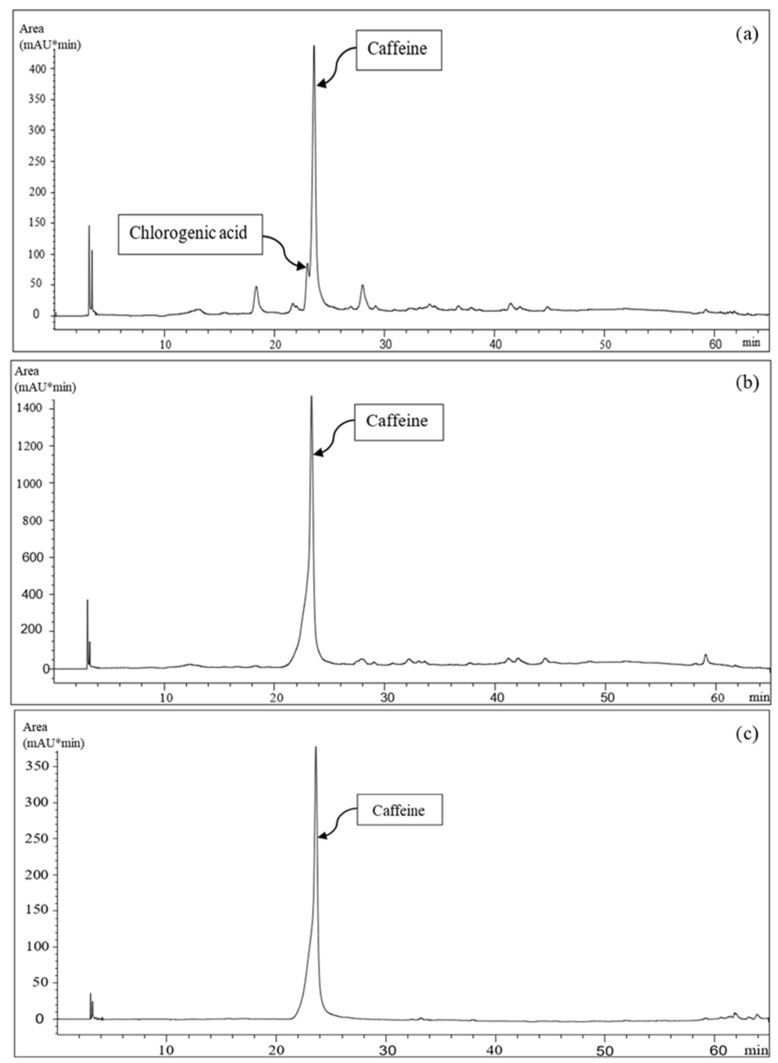
Chromatograms of the liquid extracts obtained by (**a**) HPTE, (**b**) LLE, and (**c**) SLE.

**Figure 5 foods-12-04167-f005:**
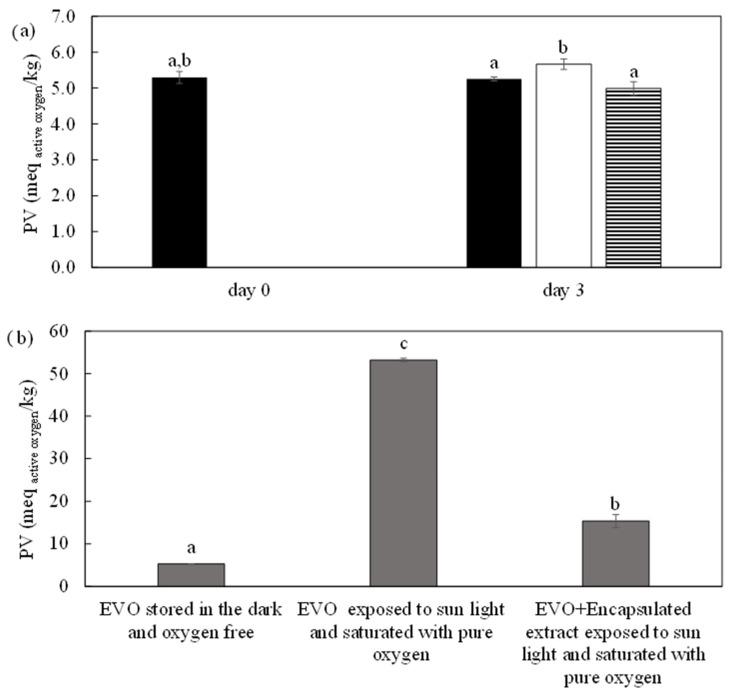
(**a**) Peroxide value (PV) of the control samples (at day 0 and after 3 days) for method validation: 
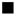
 EVO stored in the dark and oxygen free; 
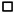
 EVO and 10% *w*/*v* wall material stored in the dark and oxygen free; 
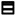
 EVO and 10% *w*/*v* encapsulated HPTE extract stored in the dark and oxygen free; (**b**) Comparison among the PV of control sample (EVO stored in the dark and oxygen free), the stressed EVO (EVO exposed to sunlight after 3 days) and the stressed EVO with the extract (EVO + encapsulated extract exposed to sunlight and saturated with pure oxygen). Different letters correspond to significant statistical differences among the data (Tukey’s post hoc test).

**Figure 6 foods-12-04167-f006:**

(**a**) Films obtained after the optimization of the PLA-based film production process (polymer-to-solvent ratio = 8% *w*/*v*, extract 0%); (**b**) film enriched with the hydroalcoholic extract (polymer-to-solvent ratio = 8% *w*/*v*, theoretical loading = 512 μg_caffeine_/g_PLA_); (**c**) film enriched with LLE extract (polymer-to-solvent ratio = 8% *w*/*v*, theoretical loading = 1494 μg_caffeine_/g_PLA_).

**Figure 7 foods-12-04167-f007:**
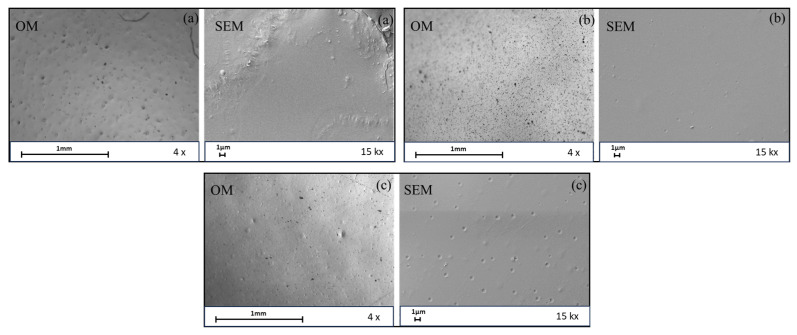
Images from an optical microscope (OM) at a 4× magnification and SEM at a 15 kx magnification of: (**a**) reference sample (0% extract), (**b**) sample enriched with the HPTE extract (512 μg _caffeine_/g _PLA_), (**c**) sample enriched with LLE extract (1494 μg _caffeine_/g _PLA_).

**Figure 8 foods-12-04167-f008:**
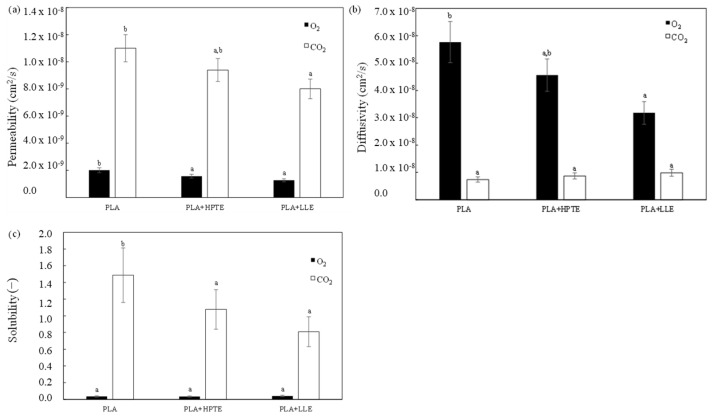
(**a**) Permeabilities, (**b**) diffusivities, and (**c**) solubilities of O_2_ and CO_2_ through the blank sample and those enriched with the hydroalcoholic extract (HPTE) and the extract further purified (LLE), with caffeine contents of 512 μg _caffeine_/g _PLA_ and 1494 μg _caffeine_/g _PLA_, respectively. Different letters within the same data series correspond to significant statistical differences among the data (Tukey’s post hoc test).

**Figure 9 foods-12-04167-f009:**
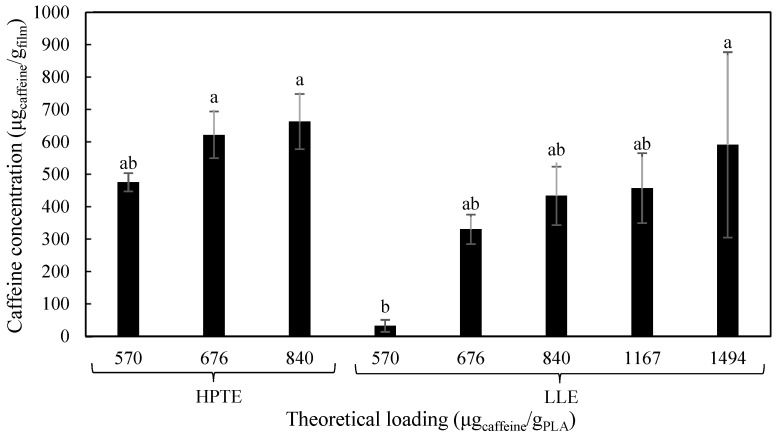
Caffeine migration at 4 °C and in 10% ethanol (*v*/*v*) as food simulant. HPTE = extract by high-pressure and -temperature extraction; LLE = extract by HPTE further purified by liquid–liquid extraction. Different letters correspond to significant statistical differences among the data (Tukey’s post hoc test).

**Figure 10 foods-12-04167-f010:**
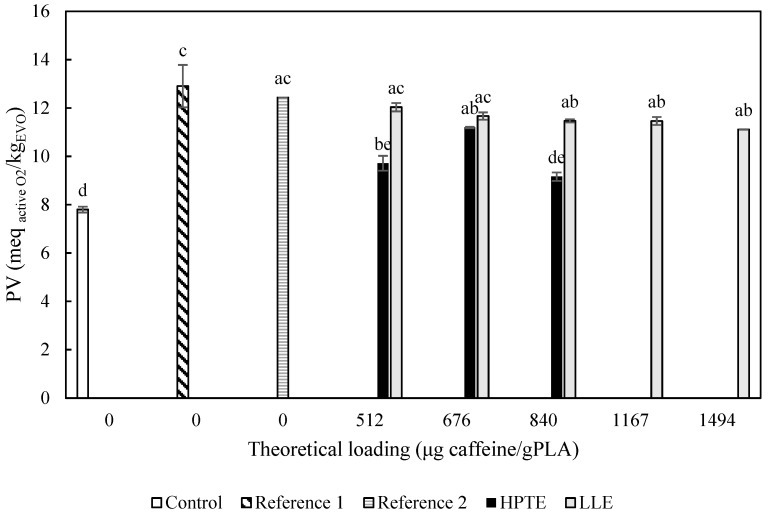
Peroxide value of EVO samples exposed to UV light for three days at room temperature. Control = EVO stored in the dark; Reference 1 = EVO exposed to UV light; reference = EVO exposed to UV light and put in contact with the reference film (42.4 cm^2^); HPTE = EVO exposed to UV light and put in contact with the films (42.4 cm^2^) loaded with HPTE extract; LLE = EVO exposed to UV light and put in contact with the films (42.4 cm^2^) loaded with HPTE extract. Different letters correspond to significant statistical differences among the data (Tukey’s post hoc test).

**Figure 11 foods-12-04167-f011:**
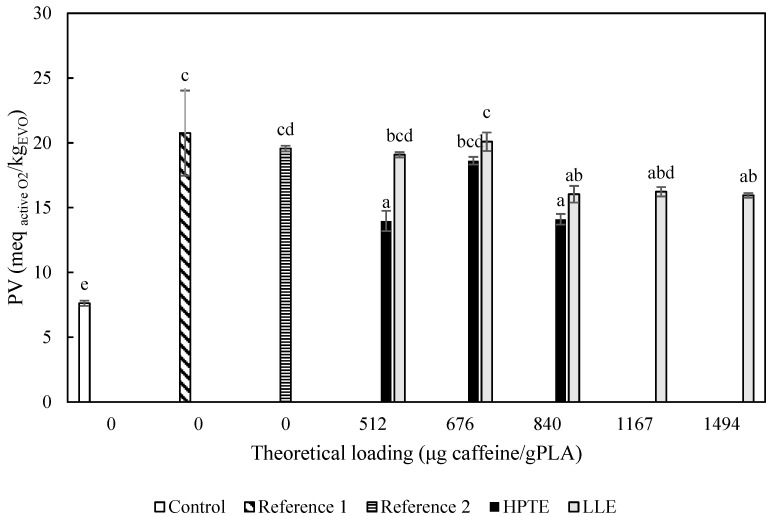
Peroxide value of EVO samples exposed to UV light for three days and sunlight for a further three days, at room temperature. Control = EVO stored in the dark; Reference 1 = EVO exposed to UV light; Reference 2 = EVO exposed to UV light and put in contact with the reference film (42.4 cm^2^); HPTE = EVO exposed to UV light and put in contact with the films (42.4 cm^2^) loaded with HPTE extract; LLE = EVO exposed to UV light and put in contact with the films (42.4 cm^2^) loaded with HPTE extract. Different letters correspond to significant statistical differences among the data (Tukey’s post hoc test).

**Table 1 foods-12-04167-t001:** Characterization of extracts obtained by high-pressure and -temperature extraction (HPTE), purification by liquid–liquid extraction (LLE), solid–liquid extraction (SLE). TE = Trolox equivalents, CAE = caffeic acid equivalents, SCG = dried spent coffee grounds. Different letters within the same column correspond to significant statistical differences among the data (Tukey’s post hoc test).

	Extract from HPTE	Extract from LLE	Extract from SLE
Extract total solids (mg _solids_/g _SCG_)	276 ± 12.6 ^a^	48.3 ± 6.7 ^b^	103.3 ± 11.7 ^c^
Antiradical power (μg _TE_/g _SCG_)	0.57 ± 0.06 ^a^	0.17 ± 0.008 ^b^	0.01 ± 0.001 ^c^
Total polyphenols (mg _CAE_/g _SCG_)	36.4 ± 1	n.d.*	n.d.*
Caffeine (mg_caffeine_/g _SCG_)	10.3 ± 0.18 ^a^	7.8 ± 0.55 ^b^	1.4 ± 0.007 ^c^
Chlorogenic acid (mg _chlorogenic acid_/g _SCG_)	2.4 ± 0.06	n.d.*	n.d.*

* n.d.: not detected.

**Table 2 foods-12-04167-t002:** Values of ARP (μg _TE_/kg _film_) over time obtained during the release test in 10% ethanol (*v*/*v*). Different letters within the same column correspond to significant statistical differences among the data (Tukey’s post hoc test).

Time (h)	Theoretical Loading HPTE Extract			Theoretical Loading LLE Extract		
	512 μg _caffeine_/g _PLA_	676 μg _caffeine_/g _PLA_	840 μg _caffeine_/g _PLA_	512 μg _caffeine_/g _PLA_	676 μg _caffeine_/g _PLA_	840 μg _caffeine_/g _PLA_
1	75 ± 10 ^ab^	60 ± 9 ^a^	37 ± 3 ^a^	26 ± 6 ^ab^	28 ± 3 ^ab^	26 ± 2 ^a^
2	72 ± 13 ^ab^	58 ± 2 ^a^	53 ± 1 ^ab^	34 ± 5 ^b^	27 ±1 ^ab^	34 ± 3 ^b^
6	81 ± 7 ^b^	45 ± 9 ^a^	61 ± 14 ^b^	19 ± 0.4 ^a^	21 ± 3 ^a^	22 ± 5 ^a^
24	65 ± 13 ^ab^	55 ± 10 ^a^	26 ± 11^c^	48 ± 5 ^c^	40 ± 9 ^b^	28 ± 2 ^ab^
30	51 ± 11 ^a^	46 ± 3 ^a^	42 ± 3 ^abc^	22 ± 3 ^a^	33 ± 10 ^ab^	28 ± 1 ^ab^
48	56 ± 3 ^ab^	57 ± 4 ^a^	53 ± 2 ^ab^	29 ± 4 ^ab^	37 ± 3 ^ab^	28 ± 3 ^ab^

## Data Availability

The data used to support the findings of this study can be made available by the corresponding author upon request.

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
