# Peer review of "Potential Use of PLA-Based Films Loaded with Antioxidant Agents from Spent Coffee Grounds for Preservation of Refrigerated Foods"

_foods, 2023, doi:10.3390/foods12224167_

Round 1

Reviewer 1 Report

Comments and Suggestions for Authors

Manuscript ID: foods-2709765

Top of Form

The manuscript reports on the development of active food packaging suitable for refrigerated foods. Polylactic acid-based films were created using the solvent casting technique, and extracts from spent coffee grounds were employed as active agents. Two types of extracts were tested: one obtained through high-pressure and temperature extraction (HPTE), and another that underwent further purification via liquid-liquid extraction (LLE). These extracts were assessed for their effects on packaging properties and the migration of active compounds in food simulants.

1.      In my opinion, the title should be “Potential use of PLA-based films loaded with antioxidant agents from spent coffee grounds in the preservation of refrigerated foods” because the authors only used food simulants as the test subjects, not actual food samples.

2.      In the Materials and Methods section (page 3; lines 99-100), please provide the molecular weight of polylactic acid. This information is valuable for readers conducting further studies.

3.  The data presentation needs improvement. Statistical analysis has been performed for Table 2, but no statistical analysis has been conducted for Table 1.

4.      The data presentation in Table 1 is unclear. What does (-) signify? It seems that Total polyphenols and Chlorogenic acid were not detected in the Extract from LLE and the Extract from SLE. Please include explanations in the footnote of the picture caption.  

5.      It is recommended to complete the statistical analysis for the data in all the figures. While statistical analysis has been performed for Figures 8 to 10, none has been conducted for Figures 4 and 7.

6.      Future works suggested in this paper: Investigating the potential impact of the obtained films on the sensory properties, such as taste and texture, of the preserved foods would provide valuable insights for practical applications.

Comments on the Quality of English Language

Author Response

We thank the reviewer for the valuable suggestion and the careful reading of the paper. All the suggestions were accepted and point by point answers are reported in the following. The modifications in the manuscript are yellow-highlighted.  

The manuscript reports on the development of active food packaging suitable for refrigerated foods. Polylactic acid-based films were created using the solvent casting technique, and extracts from spent coffee grounds were employed as active agents. Two types of extracts were tested: one obtained through high-pressure and temperature extraction (HPTE), and another that underwent further purification via liquid-liquid extraction (LLE). These extracts were assessed for their effects on packaging properties and the migration of active compounds in food simulants.

  1. In my opinion, the title should be “Potential use of PLA-based films loaded with antioxidant agents from spent coffee grounds in the preservation of refrigerated foods” because the authors only used food simulants as the test subjects, not actual food samples.

As suggested by Reviewer, paper title was modified: “Potential use of PLA-based films loaded with antioxidant agents from spent coffee grounds for preservation of refrigerated foods” 

  1. In the Materials and Methods section (page 3; lines 99-100), please provide the molecular weight of polylactic acid. This information is valuable for readers conducting further studies.

The reviewer is right. The Polymer used is a commercial PLA, Ingeo™ Biopolymer 2003D, NatureWorks. The technical specifications of this polymer are reported in the datasheet available in the web (https://www.natureworksllc.com/~/media/Technical_Resources/Technical_Data_Sheets/TechnicalDataSheet_2003D_FFP-FSW_pdf.pdf).

The data presentation needs improvement. Statistical analysis has been performed for Table 2, but no statistical analysis has been conducted for Table 1.

Authors thank the reviewer for the suggestion. The statistical analysis was added to the results in Table 1, as suggested by Reviewer.

The data presentation in Table 1 is unclear. What does (-) signify? It seems that Total polyphenols and Chlorogenic acid were not detected in the Extract from LLE and the Extract from SLE. Please include explanations in the footnote of the picture caption.  

In Table 1, the symbol (-) corresponds to” not detected”. The symbol in the table was replaced with n.d. and explanation of the symbol was included in the footnote of the Table caption, as suggested by the Reviewer.

 It is recommended to complete the statistical analysis for the data in all the figures. While statistical analysis has been performed for Figures 8 to 10, none has been conducted for Figures 4 and 7.

The statistical analysis was added to the results in the Figures 5 now numbered as 5 and 8, as suggested by Reviewer.

  1. Future works suggested in this paper: Investigating the potential impact of the obtained films on the sensory properties, such as taste and texture, of the preserved foods would provide valuable insights for practical applications.

As suggested by Reviewer, a final sentence about future perspectives was added, in particular: “Among future perspectives of this research work, the investigation of the potential impact of the obtained films on the food sensory properties, such as taste and texture, is planned. Indeed, it could be a valuable insight to individuate the most suitable applications of the produced active packaging”.

Reviewer 2 Report

Comments and Suggestions for Authors

L65 : PLA is not hydrophobic since studies have shown that contact angle with water drop is lower than 90°
L71 : casting is not the most important process ability, since it is thermoplastic. Casting is a possibility, but using hazardous solvent. Please change this sentence according to my remark
L72 : write CO2 with « 2 » underlined
L320 : indicate the thickness of the films according to the ratio PLA/Solvent
L346 : add a figure showing HPLC Chromatograms of the 3 extracts so readers can see the difference in their composition.
L469 : Indicate scales on every SEM and OM images
L510 : in the graphs, write CO2 and O2 with the « 2 » in subscript
L 619 : and other references : please indicate the page

Comments on the Quality of English Language

Minor editing of English language required

Author Response

We thank the reviewer for the valuable suggestion and the careful reading of the paper. All the suggestions were accepted and point by point answers are reported in the following. The modifications in the manuscript are yellow-highlighted. 

L65 : PLA is not hydrophobic since studies have shown that contact angle with water drop is lower than 90°

As suggested by Reviewer, we modified the line 65 removing the term “hydrophobic” in the description of the polymer.

L71 : casting is not the most important process ability, since it is thermoplastic. Casting is a possibility, but using hazardous solvent. Please change this sentence according to my remark

As suggested by Reviewer, the text at line 72 was modified specifying that solvent casting is a possible processing technology for polylactic acid, but it could entail the use of hazardous solvents, like chloroform.

L72 : write CO2 with « 2 » underlined

At line 74, CO2 was corrected with CO2.

L320 : indicate the thickness of the films according to the ratio PLA/Solvent

According to reviewer suggestions, at line 334, a sentence was added in the revised manuscript: “It did not significantly affected the final film thickness, that was comprised between 40 and 50 µm at the conditions tested”.

L346 : add a figure showing HPLC Chromatograms of the 3 extracts so readers can see the difference in their composition.

As suggested by Reviewer, the HPLC chromatograms of the extracts were added as Figure 4.

L469 : Indicate scales on every SEM and OM images

Scales were added on the figures. Furthermore, the magnifications set for OM and SEM (4x and 15 kx, respectively) were inserted as additional information in the figure’s caption  (L463-497).

L510 : in the graphs, write CO2 and O2 with the « 2 » in subscript

Changes required by the reviewer have been performed.

L 619 : and other references : please indicate the page

Authors apologize for the oversights. The reference list was corrected with the information required.